## [Reviewer comments · BMJ Open]

ARTICLE DETAILS

TITLE (PROVISIONAL)	'That's how we got around it': A qualitative exploration of healthcare professionals' experiences of care provision for asylum applicants' with Limited English Proficiency in UK contingency accommodation
AUTHORS	Tomkow, Louise; Prager, Gabrielle; Drinkwater, Jessica; Morris, Rebecca L.; Farrington, Rebecca

VERSION 1 – REVIEW

REVIEWER	Pottie, Kevin Schulich School of Medicine and Dentistry, Family Medicine
REVIEW RETURNED	26-May-2023

GENERAL COMMENTS	Extremely well written and interesting paper and important topic. I appreciate the value of viewing the risk for discrimination with LEP patients. I believe the paper could be stronger with: More information regarding the existing interpreter service- time to access, quality of interpretation service- these are important elements. The authors did mention the competing interests that may reduce the use of interpreter service, ie need to see many patients. I think it is also worth mentioning that the income of the practitioner or work hours could be affected and this could also contribute to lack of use of interpreters. The practitioners must be trained to work with interpreters, and the system must work efficiently to reduce other contributing factors to lack of use of interpreters. I believe the discussion should mention a few more US references re: the use of term language discrimination. How does it compare to present UK study. The limitation of number of participants is clearly stated, however I recommend not to use the quantitative research term of 'sample size' and rather to speak of small or modest number of participants. The authors could consider mention of universal access to quality care to contextualize the importance of interpreters. The authors could, in discussion section, mention potential relevance to trauma informed care for vulnerable populations.
--

REVIEWER	Kienzler, Hanna King's College London
REVIEW RETURNED	13-Jun-2023

GENERAL COMMENTS	The authors address a timely topic exploring, from the perspective of health providers, how inadequate language provision to people with Limited English Proficiency (LEP) can be a determinant of
--

poor health. Qualitative research was conducted with 10 NHS staff serving refugees living in contingency housing during the Covid-19 pandemic. Results highlighted high level of LEP among refugee patients and a lack of interpreters to bridge the language divide. Interestingly, staff did not challenge this gap as discriminatory; nevertheless, they developed workarounds to deal with the situation. Such workarounds, it is argued, can lead to short cuts and possibly inadequate care. While the paper tackles an important topic, it needs to be more nuanced and strengthened conceptually. Recommendations for improvement are below.

Background

(1) It would have been useful to provide a definition asylum seekers and a clearer outline of their rights, especially those pertaining to healthcare, in the context of the UK. See for example: Asif, Z., & Kienzler, H. (2022). Structural barriers to refugee, asylum seeker and undocumented migrant healthcare access. *Perceptions of Doctors of the World caseworkers in the UK. SSM-Mental Health*, 2, 100088.

Pollard, T., & Howard, N. (2021). Mental healthcare for asylum-seekers and refugees residing in the United Kingdom: a scoping review of policies, barriers, and enablers. *International Journal of Mental Health Systems*, 15(1), 1-15.

(2) While the importance of language interpretation in the context of healthcare is outlined, information is lacking on the current situation in the UK. What language interpretation services are provided in the NHS? If language interpretation is provided, what languages are generally covered? Where are these services mostly provided and where are they lacking? What role do charities, the voluntary sector and family members play in this field? What are some of the downsides of the use of language interpreters especially among small groups (e.g., related to trust, privacy and confidentiality)?

Theoretical framework:

Overall, the paper needs to be conceptually stronger.

(1) The authors need to better explain when and how a lack of language interpretation translates into 'discrimination' and, more concretely, 'language discrimination'.

(2) Depending how the authors unpack the lack of language interpretation as a form of discrimination, the latter might be one of several forms of structural violence that affect the refugee community in the current political 'hostile environment' of the UK. This broader context of discrimination/systemic oppression needs to be outlined explicitly both empirically and conceptually in order to provide the context within which language discrimination takes place.

(3) To what extent does the call for language interpretation link with notions of cultural competence or, preferred, cultural humility? It might be important to engage with this set of literature as it could be argued that translation without awareness of cultural expressions of distress and ill health does not go far enough in the provision of adequate healthcare for asylum seekers.

Methods:

While I accept that no personal information can be shared about participants to safeguard anonymity, I do believe that it would be important (and possible) to share some demographic information (gender and age distribution), qualifications, length of time

providing healthcare to asylum seekers, geographical location. This information does not have to be matched to individual participants. However, it is important to know the positionality from which the expert experiences are shared and where, in the UK, the experiences were had (healthcare provision and access to healthcare for refugees differs quite significantly according location).

Results:

(1) The title for sub-header (ii) is somewhat awkwardly phrased. I do not assume that this is an article about the 'importance of language,' but rather one about the importance of language translation or of speaking the same language. This opens another question on language – to what extent do the authors account for the fact that several health providers are multi-lingual and can bridge some of the language gaps? That is, to what extent are the diverse cultural backgrounds among healthcare staff taken into consideration? This is not to take away from the fact that language translation is clearly missing and problematic in the NHS; it is just to highlight that diversity is not just represented in the patient population but also in the staff population which can be an asset that should not be ignored.

(2) In section (ii), more information could have been provided on what it takes to provide healthcare when patients and practitioners do not understand each other. How do practitioners assess language proficiency? How do they respond to different levels of such proficiency (i.e., I am looking for more nuance considering that patients do not either have good English or no English proficiency – there is a wide range in between).

(3) The quote on page 8 suggests that language translation needs to come in different forms besides having interpreters available (e.g., translation of signage, pamphlets and labels into different languages).

(4) In section (iii) the mention of the use of community volunteers is interesting. In the discussion, this would need to be critically assessed in relation to existing literature that highlights problems related to trust and privacy and confidentiality among small communities.

(5) The information on 'work arounds' is interesting and fits with literature on 'tinkering' in the clinical space where there is a lack of resources and standards of care. I would urge the authors to familiarise themselves with this work and maybe give the health providers more credit for the creativity they display in the face of scarce resources and systemic inequity directed at their patients. More could have been made of the section on 'what is it that health providers do' in the restrictive environment in which they are forced to work. What can be learned from them?

(6) I did not find that the quotes provided suggested 'normalisation' of the lack of language interpretation. Might the authors unjustly judge health providers as they are trying to provide care in very difficult circumstances and are, in fact, willing to experiment and 'tinker' to provide the best care they can? This does not mean that the services, thus, provided are adequate; but, it would at least recognise the work that is being done to provide some form of support and care.

Discussion:

The discussion needs to be more nuanced and could pick up some of the concepts I suggest above to highlight what it is that practitioners do in the context of scarcity and systemic inequity. If

	the takeaway message is 'more translators in the health sector' then, this does not go far enough considering that the lack of translation is but one aspect of a more complex discriminatory system at play. The changes need to be systemic and not merely technical (although these are also needed).
--	--

VERSION 1 – AUTHOR RESPONSE

Reviewer: 1

Dr. Kevin Pottie, Schulich School of Medicine and Dentistry

Comments to the Author:

Extremely well written and interesting paper and important topic.

I appreciate the value of viewing the risk for discrimination with LEP patients.

I believe the paper could be stronger with:

- More information regarding the existing interpreter service- time to access, quality of interpretation service- these are important elements.

Extra information about this has been added to the 'research context' section.

- The authors did mention the competing interests that may reduce the use of interpreter service, ie need to see many patients. I think it is also worth mentioning that the income of the practitioner or work hours could be affected and this could also contribute to lack of use of interpreters. The practitioners must be trained to work with interpreters, and the system must work efficiently to reduce other contributing factors to lack of use of interpreters.

Many thanks, we have added a reference to the training and integration needed to the first paragraph of the conclusion section.

- I believe the discussion should mention a few more US references re: the use of term language discrimination. How does it compare to present UK study.

Many thanks for this suggestion. We have added a brief summary overview of the current state of the literature in the US and the unique contribution our work makes.

- The limitation of number of participants is clearly stated, however I recommend not to use the quantitative research term of 'sample size' and rather to speak of small or modest number of participants.

Many thanks, we have changed this

- The authors could consider mention of universal access to quality care to contextualize the importance of interpreters.

We hope that the first three paragraphs in the introduction set out the importance of this, however we had added the term 'universal access' to further clarify the point. Many thanks for this suggestion.

- The authors could, in discussion section, mention potential relevance to trauma informed care for vulnerable populations.

Many thanks, we have added a reference to trauma-informed care in the first paragraph of the discussion section.

Reviewer: 2

Dr. Hanna Kienzler, King's College London

Comments to the Author:

The authors address a timely topic exploring, from the perspective of health providers, how inadequate language provision to people with Limited English Proficiency (LEP) can be a determinant

of poor health. Qualitative research was conducted with 10 NHS staff serving refugees living in contingency housing during the Covid-19 pandemic. Results highlighted high level of LEP among refugee patients and a lack of interpreters to bridge the language divide. Interestingly, staff did not challenge this gap as discriminatory; nevertheless, they developed workarounds to deal with the situation. Such workarounds, it is argued, can lead to short cuts and possibly inadequate care. While the paper tackles an important topic, it needs to be more nuanced and strengthened conceptually. Recommendations for improvement are below.

Many thanks for your detailed guidance and suggestions. We have carefully considered each point and have made major revisions to the paper based on your feedback. This has been significantly useful in two main areas:

- your pushing us towards speaking about the context in which this discrimination operates has encouraged a more rounded and critical analysis of the conditions which face migrants in the UK
- your suggestion we examine the concept of tinkering, which has allowed us to reframe our analysis of the practice of the staff.

As you will see, we have been unable to adopt every suggestion, due the focus and scope of the data we collected and due to the word count of the journal. Where we have been unable to do this we have indicated in the responses below. Nevertheless, we thank you for your comments and feel your feedback has greatly strengthened the paper by refocussing the critique away from individual practitioners towards the governmental hostile environment.

Background

(1) It would have been useful to provide a definition asylum seekers and a clearer outline of their rights, especially those pertaining to healthcare, in the context of the UK. See for example: Asif, Z., & Kienzler, H. (2022). Structural barriers to refugee, asylum seeker and undocumented migrant healthcare access. Perceptions of Doctors of the World caseworkers in the UK. *SSM-Mental Health*, 2, 100088.

Pollard, T., & Howard, N. (2021). Mental healthcare for asylum-seekers and refugees residing in the United Kingdom: a scoping review of policies, barriers, and enablers. *International Journal of Mental Health Systems*, 15(1), 1-15.

Many thanks for this suggestion, we have added detail here using the UNHCR definitions and added some more detail about the UK asylum system.

(2) While the importance of language interpretation in the context of healthcare is outlined, information is lacking on the current situation in the UK. What language interpretation services are provided in the NHS? If language interpretation is provided, what languages are generally covered? Where are these services mostly provided and where are they lacking? What role do charities, the voluntary sector and family members play in this field? What are some of the downsides of the use of language interpreters especially among small groups (e.g., related to trust, privacy, and confidentiality)?

Many thanks for this suggestion. We have added a short description of the current state of things in the UK, supported by some key references. Due to the restrictions of the word count we have kept this brief, outlining that the provision is highly variable and frequently described by service users as inadequate. The more specific answers to these questions can be found in the references, for readers who want to understand more.

- Care providers have a duty to provide interpretation services in line with the needs of the patient.

This is not dependent on language spoken. The population the area we were researching is superdiverse, >220 languages spoken.

- As described in the paper, during the pandemic third sector organisations stepped into this service, and usually family members play a key role in interpretation, with ethical concerns.

- The Barron et al paper contains a more comprehensive overview of the nuances of interpretation for

small diaspora groups where confidentiality may be an issue within communities. We feel it is beyond the scope of this paper to unpick this fully unfortunately due to the word limit, but acknowledge this can be an issue.

- <https://www.england.nhs.uk/wp-content/uploads/2018/09/guidance-for-commissioners-interpreting-and-translation-services-in-primary-care.pdf>
- Kang, C., Tomkow, L. and Farrington, R., 2019. Access to primary health care for asylum seekers and refugees: a qualitative study of service user experiences in the UK. *British Journal of General Practice*, 69(685), pp.e537-e545.
- Barron, D.S., Holterman, C., Shipster, P., Batson, S. and Alam, M., 2010. Seen but not heard—ethnic minorities' views of primary health care interpreting provision: a focus group study. *Primary Health Care Research & Development*, 11(2), pp.132-141.
- Gill, P.S., Beavan, J., Calvert, M. and Freemantle, N., 2011. The unmet need for interpreting provision in UK primary care. *PLoS one*, 6(6), p.e20837.

Theoretical framework:

Overall, the paper needs to be conceptually stronger.

(1) The authors need to better explain when and how a lack of language interpretation translates into 'discrimination' and, more concretely, 'language discrimination'.

Many thanks for this. We have refined our theoretical framework section and expanded our conceptualisation of Language Discrimination, adding: We conceptualise language discrimination as both structural – built into and normalised in mainstream institutions - and structuring – in that it reinforces existing hierarchies of privilege, restricting the opportunities of already-disadvantaged groups. The inadequate provision of interpretation services epitomises the operationalisation of language discrimination in healthcare and is currently entangled in the current UK context of a Hostile Environment. It is important to note that adequate provision of interpreters would not eradicate language discrimination as cultural competency remains important – however, for the purpose of this paper we use adequate interpretation provision as the bare minimum, thus a proxy marker of language discrimination, utilising it as an empirical signal through which we explore the issue.

(2) Depending how the authors unpack the lack of language interpretation as a form of discrimination, the latter might be one of several forms of structural violence that affect the refugee community in the current political 'hostile environment' of the UK. This broader context of discrimination/systemic oppression needs to be outlined explicitly both empirically and conceptually in order to provide the context within which language discrimination takes place.

Many thanks for this suggestion. We have added further detail about the hostile environment in the introduction and research context and have made the connection to this in the theoretical framework (above) and discussion section also

(3) To what extent does the call for language interpretation link with notions of cultural competence or, preferred, cultural humility? It might be important to engage with this set of literature as it could be argued that translation without awareness of cultural expressions of distress and ill health does not go far enough in the provision of adequate healthcare for asylum seekers.

The issue of cultural humility is an important one as cultural humility is of course an important part of a form of patient centredness. We feel that, for the purposes of this paper we have inadequate data and space to address this adequately. We have clarified that, whilst even with adequate interpretation provision, discrimination, and a lack of cultural humility in consultations and care may still occur. For the purposes of this paper we are examining interpretation, not as a gold standard, but as a bare minimum, and using it as a proxy gauge for language discrimination. We hope this is clearer in the revised version.

Methods:

While I accept that no personal information can be shared about participants to safeguard anonymity, I do believe that it would be important (and possible) to share some demographic information (gender and age distribution), qualifications, length of time providing healthcare to asylum seekers, geographical location. This information does not have to be matched to individual participants. However, it is important to know the positionality from which the expert experiences are shared and where, in the UK, the experiences were had (healthcare provision and access to healthcare for refugees differs quite significantly according to location).

Thank you for this comment. We have taken time to consider this feedback. We understand that in theory this would be possible, however, we as authors will be easily geographically identifiable once this work is published. There is only one such specialist service in the local area and a very small team with only one or two individuals in particular roles. The staff and residents have been targeted by right wing groups during the pandemic. We take the anonymity of the participants very seriously and feel, as a team, that we are uncomfortable with providing any further information about the individuals who volunteered to share their experiences.

Results:

(1) The title for sub-header (ii) is somewhat awkwardly phrased. I do not assume that this is an article about the 'importance of language,' but rather one about the importance of language translation or of speaking the same language. This opens another question on language – to what extent do the authors account for the fact that several health providers are multi-lingual and can bridge some of the language gaps? That is, to what extent are the diverse cultural backgrounds among healthcare staff taken into consideration? This is not to take away from the fact that language translation is clearly missing and problematic in the NHS; it is just to highlight that diversity is not just represented in the patient population but also in the staff population which can be an asset that should not be ignored. Many thanks for highlighting this. We have amended it to read: The importance of interpretation provision for safe care. We did not collect data on the language skills of the healthcare staff and so exploration of this is beyond the scope of this paper, though we do recognise it as an issue worthy of exploration.

(2) In section (ii), more information could have been provided on what it takes to provide healthcare when patients and practitioners do not understand each other. How do practitioners assess language proficiency? How do they respond to different levels of such proficiency (i.e., I am looking for more nuance considering that patients do not either have good English or no English proficiency – there is a wide range in between).

Thankyou for this comment, unfortunately, we did not collect data on how practitioners assess language proficiency and feel unable to comment on this in our analysis.

(3) The quote on page 8 suggests that language translation needs to come in different forms besides having interpreters available (e.g., translation of signage, pamphlets and labels into different languages).

Many thanks for this. We have added further detail to highlight this important point.

(4) In section (iii) the mention of the use of community volunteers is interesting. In the discussion, this would need to be critically assessed in relation to existing literature that highlights problems related to trust and privacy and confidentiality among small communities.

Again, as our data did not suggest significant tensions within the provision from community volunteers, we feel this is beyond this scope and focus of this study.

(5) The information on 'work arounds' is interesting and fits with literature on 'tinkering' in the clinical space where there is a lack of resources and standards of care. I would urge the authors to familiarise themselves with this work and maybe give the health providers more credit for the creativity they

display in the face of scarce resources and systemic inequity directed at their patients. More could have been made of the section on 'what is it that health providers do' in the restrictive environment in which they are forced to work. What can be learned from them?

Many thanks for highlighting this literature to us, which was extremely valuable. As you will see, we have used it to reframe the paper considerably. This has helped us re-examine the innovative work arounds undertaken by the staff and, as you suggested, focus on the value added by these innovations.

(6) I did not find that the quotes provided suggested 'normalisation' of the lack of language interpretation. Might the authors unjustly judge health providers as they are trying to provide care in very difficult circumstances and are, in fact, willing to experiment and 'tinker' to provide the best care they can? This does not mean that the services, thus, provided are adequate; but, it would at least recognise the work that is being done to provide some form of support and care.

We hope that though reframing the analysis using the concept of tinkering we have addressed this excellent suggestion. The last thing we wanted to do was unjustly judge the staff interviewed and we feel the paper is much stronger for the introduction of the concept of tinkering.

Discussion:

The discussion needs to be more nuanced and could pick up some of the concepts I suggest above to highlight what it is that practitioners do in the context of scarcity and systemic inequity. If the takeaway message is 'more translators in the health sector' then, this does not go far enough considering that the lack of translation is but one aspect of a more complex discriminatory system at play. The changes need to be systemic and not merely technical (although these are also needed).

We have rewritten the discussion, informed by the feedback above. Specifically, we have:

- An reflection on the concept of tinkering, and the ways in which this can unintentionally uphold imperfect systems
- Increased our focus on the broader context, and recognition – as you say – that the issue of language discrimination is merely one facet of a highly hostile environment facing people who migrate to the UK
- Reflected this in our recommendations